# Who Are More Likely to Have Quit Intentions among Malaysian Adult Smokers? Findings from the 2020 ITC Malaysia Survey

**DOI:** 10.3390/ijerph19053035

**Published:** 2022-03-04

**Authors:** Siti Idayu Hasan, Susan C. Kaai, Amer Siddiq Amer Nordin, Farizah Mohd Hairi, Mahmoud Danaee, Anne Yee, Nur Amani Ahmad Tajuddin, Ina Sharyn Kamaludin, Matt Grey, Mi Yan, Pete Driezen, Mary E. Thompson, Anne C. K. Quah, Geoffrey T. Fong

**Affiliations:** 1Nicotine Addiction Research Group, University of Malaya Centre of Addiction Sciences, University of Malaya, Kuala Lumpur 50603, Malaysia; ayu_umcas@um.edu.my (S.I.H.); farizah@um.edu.my (F.M.H.); mdanaee@um.edu.my (M.D.); annyee17@um.edu.my (A.Y.); amaninatasha@um.edu.my (N.A.A.T.); inakamaludin@um.edu.my (I.S.K.); 2Department of Psychology, University of Waterloo, 200 University Ave. W., Waterloo, ON N2L 3G1, Canada; skaai@uwaterloo.ca (S.C.K.); mgrey@uwaterloo.ca (M.G.); mi.yan@uwaterloo.ca (M.Y.); prdriezen@uwaterloo.ca (P.D.); ackquah@uwaterloo.ca (A.C.K.Q.); gfong@uwaterloo.ca (G.T.F.); 3School of Public Health Sciences, University of Waterloo, 200 University Ave. W., Waterloo, ON N2L 3G1, Canada; 4Department of Psychological Medicine, Faculty of Medicine, University of Malaya, Kuala Lumpur 50603, Malaysia; 5Department of Social and Preventive Medicine, Faculty of Medicine, University of Malaya, Kuala Lumpur 50603, Malaysia; 6Department of Primary Care Medicine, Faculty of Medicine, University of Malaya, Kuala Lumpur 50603, Malaysia; 7Department of Statistics and Actuarial Sciences, University of Waterloo, 200 University Ave. W., Waterloo, ON N2L 3G1, Canada; methompson@uwaterloo.ca; 8Ontario Institute for Cancer Research, 661 University Ave., Suite 510, Toronto, ON M5G 0A3, Canada

**Keywords:** quit intentions, smoking, predictors, cessation, Malaysia

## Abstract

Increasing quitting among smokers is essential to reduce the population burden of smoking-related diseases. Smokers’ intentions to quit smoking are among the strongest predictors of future quit attempts. It is therefore important to understand factors associated with intentions to quit, and this is particularly important in low- and middle-income countries, where there have been few studies on quit intentions. The present study was conducted to identify factors associated with quit intentions among smokers in Malaysia. Data came from the 2020 International Tobacco Control (ITC) Malaysia Survey, a self-administered online survey of 1047 adult (18+) Malaysian smokers. Smokers who reported that they planned to quit smoking in the next month, within the next six months, or sometime beyond six months were classified as having intentions to quit smoking. Factors associated with quit intentions were examined by using multivariable logistic regression. Most smokers (85.2%) intended to quit smoking. Smokers were more likely to have quit intentions if they were of Malay ethnicity vs. other ethnicities (adjusted odds ratio (AOR) = 1.82, 95% confidence interval (CI) = 1.03–3.20), of moderate (AOR = 2.11, 95% CI = 1.12–3.99) or high level of education vs. low level of education (AOR = 1.97, 95% CI = 1.04–3.75), if they had ever tried to quit smoking vs. no quit attempt (AOR = 8.81, 95% CI = 5.09–15.27), if they received advice to quit from a healthcare provider vs. not receiving any quit advice (AOR = 3.78, 95% CI = 1.62–8.83), and if they reported worrying about future health because of smoking (AOR = 3.11, 95% CI = 1.35–7.15 (a little worried/moderately worried vs. not worried); AOR = 7.35, 95% CI = 2.47–21.83 (very worried vs. not worried)). The factors associated with intentions to quit smoking among Malaysian smokers were consistent with those identified in other countries. A better understanding of the factors influencing intentions to quit can strengthen existing cessation programs and guide the development of more effective smoking-cessation programs in Malaysia.

## 1. Introduction

Tobacco smoking is one of the leading causes of preventable death and disease globally and remains one of the biggest public health threats worldwide. One in 10 adult deaths is caused by smoking every year, equating to more than eight million deaths worldwide annually [1]. A total of 7.1 million of those deaths are directly attributable to tobacco smoking, while an additional 1.2 million result from exposure to secondhand smoke [1]. Smoking is more prevalent in low- and middle-income countries (LMICs), where more than 80% of the world’s 1.1 billion smokers reside [1]. 

Over the past three decades, smoking-related diseases have been the main causes of premature death in Malaysia [2]. This figure increased to 20,000 in 2015 [3]. More recent data revealed that 27,000 Malaysians die each year due to smoking-related illnesses [4]. Malaysia faces many challenges in its effort to reduce the prevalence of smoking. The 2019 National Health and Morbidity Survey (NHMS) showed that the prevalence of smoking among Malaysians aged 15 years and older was 21.3% (i.e., 4.8 million people), with a higher percentage of smoking among males (40.5%) than females (1.2%) [5]. This was a slight reduction compared with the smoking prevalence rates reported earlier in the 2015 NHMS, i.e., 22.8% (males: 43.0% and females: 1.4%) [6]. This improvement may be attributed partly to the smoking cessation programs that were implemented in 2015.

Malaysia’s tobacco control program has included support for smoking cessation. For example, the mQuit program was developed under the National Strategic Plan on Tobacco Control 2015–2020 (NSPTC) to strengthen cessation services in Malaysia [7]. The mQuit program had several components: to expand smoking cessation clinics throughout Malaysia, to ensure the availability of cessation treatment, to update clinical practice guidelines, to improve training to provide cessation services in the country, and to set up a national Quitline [7,8]. From 2015 to 2020, the number of smokers registered under the mQuit program in both public and private clinics increased from 7757 to 28,167, and the proportion of smokers who reported quitting smoking increased from 24.0% in 2012 to 50.6% in 2020 [9]. There is also a pattern of increases in quit attempts among younger Malaysians aged 15–19 and among highly educated Malaysians, suggesting that the mQuit program has had a positive impact. 

Most smokers want to quit smoking, but nicotine dependence makes successful cessation difficult [10,11]. In many countries, while more than half of current smokers want to quit smoking, and one-third made at least three attempts in the previous year, less than half succeed in quitting before the age of 60 [12]. Several theoretical frameworks have been used to describe and understand the process—both challenges and successes—of smoking cessation, including the Transtheoretical Model (TTM) [13,14,15,16]. According to the TTM [15], individual smokers must progress through five stages of behavior change (i.e., pre-contemplation, contemplation, preparation, action, and maintenance) before they can successfully quit smoking. They begin by not thinking about any behavior change, and, at this stage, they have no plan to stop smoking; then they move to a second stage, where they seriously consider the pros and cons of quitting smoking. This is followed by preparing themselves to quit, such as setting a quit date, then implementing the plan to stop smoking, and finally maintaining this behavior to avoid relapse. Having a quit intention falls mainly in the preparation stage; thus, it is a key step to successful quitting [17,18].

Consistent with this conceptual model, studies have found that having an intention to quit smoking is one of the strongest predictors of making future attempts to quit and of successful quitting [19,20]. ITC studies in several countries have documented the strength of intentions to quit to predict future quitting behaviors in Canada, the US, the UK, Australia [21], and China [22]. For example, among Chinese smokers, among those who had intentions to quit, 41% then made at least one attempt to quit in the following year, compared to only 17% who reported having no intentions to quit [22]. An ITC cross-country comparison study between smokers in Malaysia and Thailand found that more immediate quitting intentions predicted both making a quit attempt and staying quit [23]. Other studies in China and Iran found that intentions to quit and lower daily cigarette consumption predicted sustained quitting (at 1 year follow up) among participants in a Quit and Win competition [24,25]. Because of the strong linkage between quit intentions and future quitting, identifying the factors that are associated with quit attentions offers important insights into quitting among smokers. 

Factors that are associated with quit intentions include higher perceived vulnerability to disease [23], self-efficacy [10,23,26], past quit attempts [23,27,28,29,30], nicotine dependence [28], duration of past quit attempts, outcome expectancy of quitting, worry about future health because of tobacco use [20,27,28,31], overall opinion of smoking [28,29], and subjective norms [29]. In addition, income, education [32], being male, and having a spouse [10,26] have all been shown to be predictive of quit intentions. However, few studies have examined factors associated with intentions to quit in LMICs, including Southeast Asia. Most studies in Southeast Asia have focused on quit attempts [33] and successful smoking cessation [23,34,35]. To the best of our knowledge, only one previous study has examined quitting behaviors among smokers from Malaysia [23]. Therefore, the purpose of this study was to examine the factors associated with quit intentions among current cigarette smokers from Malaysia, using data from the 2020 ITC Malaysia Wave 1 (MYS1) Survey.

## 2. Materials and Methods

### 2.1. Study Sample

The 2020 ITC MYS1 Survey was conducted among 1253 adult respondents (1047 cigarette adult smokers and 206 non-smokers) aged 18 and older. Respondents were recruited from the Rakuten Insight web panel in Malaysia that was nationally representative of Malaysian internet users (which was 88% of all Malaysians in 2020) [36]. This survey is part of the larger global ITC Project, which has conducted longitudinal cohort surveys in 30 other countries. 

The data were collected from 5 February to 3 March 2020, with an overall response rate of 11.3% and a cooperation rate of 95.3% [37,38]. It should be noted that the response rate includes in the denominator people who were invited but may never have seen or noticed the invitation. Thus, the response rate underestimates (to an unknown extent) the proportion who would have responded to an invitation received [39]. Because of this, the cooperation rate may be a better indicator of the potential sampling bias due to the content of the survey (tobacco use). The very high cooperation rate is a sign that the potential bias due to refusal on the basis of survey content was very low because eligibility was confirmed following consent to participate prior to the interview.

Respondents for this study included in statistical analyses were restricted to 1047 current cigarette smokers. Current smokers were defined as those who smoked at least monthly and had smoked 100 or more cigarettes in their lifetime. Sampling weights were computed by using a ranking algorithm and calibrated to estimated population sizes. All participants provided informed consent prior to completing the web survey, and the survey received ethical clearance from the Office of Research Ethics, University of Waterloo, Canada (ORE#40825); and the Medical Research Ethics Committee, University of Malaya (MREC ID#2019118-8018).

### 2.2. Measures

Current smokers were asked whether they were planning to quit smoking (dependent variable). Respondents reporting “within the next month”, “between 1 and 6 months”, or “sometime in the future, beyond 6 months” were classified as intending to quit, while those who responded “not planning to quit” were classified as not intending to quit. Sociodemographic variables were gender (male and female), age at the time of the survey (18–24, 25–39, 40–54, 55 years and older), highest level of education (low (upper secondary and below), moderate (diploma certificate), and high (university and higher)), and ethnicity (Malay, non-Malay (Chinese, Indian, Iban, Kadazan, Murut, Bajau, and others)). Nicotine dependence was measured by using the Heaviness of Smoking Index (HSI; 7 levels, 0 = least addicted to 6 = most addicted), which was based on the sum of two categorical variables: number of cigarettes smoked per day (CPD; 0 = 0–10, 1 = 11–20, 2 = 21–30, and 3 = 31+) and time to first cigarette (0 = >60 min, 1 = 31–60 min, 2 = 6–30 min, 3 = 5 min, or less). 

Smokers also reported whether they ever tried to quit smoking cigarettes (1 = Yes, 0 = No); worry about their health in the future because of smoking (“not at all”, “a little/moderately worried”, and “very worried”), and whether they had visited a healthcare provider and received advice to quit smoking in the last 12 months (1 = Yes, visited a healthcare provider and received advice to quit; 2 = Yes, visited a healthcare provider but did not receive advice to quit; and 0 = No, I did not visit a healthcare provider). Concern about the expense of cigarettes was assessed by asking smokers for their rating of agreement with the statement: “I spend too much money on cigarettes” (recoded into two categories: “disagree/strongly disagree/neutral” and “agree/strongly agree”).

### 2.3. Statistical Analysis

Data were analyzed by using the survey procedures in SPSS (Version 25) to account for the stratified design and sampling weights. Each independent variable was first tested in separate bivariate models to assess the association between each predictor (independent variable) and quit intentions (dependent variable). A multivariable model was then fitted to include all variables controlling for gender and age.

## 3. Results

### 3.1. Characteristics of the Sample

Most smokers were male (90.2%), Malay (52.8%), aged 39 and younger (74.4%), and reported a low or moderate level of education (62.6%) (Table 1). 

Most smokers (85.2%) reported that they intended to quit smoking sometime in the future, while 13.6% reported intending to quit within the next month, 28.4% within the next six months, and 43.2% sometime beyond six months (Table 2).

### 3.2. Factors Associated with Quit Intentions

Findings from the bivariate regression analyses (table not included) showed that smokers of moderate education vs. those with a low level of education (OR = 1.77, 95% CI = 1.04–3.03), had tried quitting smoking in the past 12 months vs. those who did not report a quit attempt (OR = 7.69, 95% CI = 4.65–12.73), visited a healthcare provider and received quitting advice vs. smokers who did not receive quitting advice from a healthcare provider (OR = 3.96, 95% CI = 1.95–8.02), were worried that smoking would damage their health in the future a little/moderately (OR = 4.64, 95% CI = 2.46–8.74), or were very worried (OR = 11.94, 95% CI = 4.44–32.10) vs. those who were not worried at all, and those concerned that they spent too much money on cigarettes vs. smokers who did not have this concern (OR = 1.96, 95% CI = 1.23–3.13) were more likely to have a quit intention. Smokers who were older (aged 55+) vs. younger smokers aged 18–24 (OR = 0.27, 95% CI = 0.10–0.77) were less likely to have quit intentions.

After adjusting for all other variables in the multivariable model (Table 3), five factors were significantly associated with quit intentions: Malay ethnicity vs. other ethnicities (AOR = 1.82, 95% CI = 1.03–3.20), having a moderate (AOR = 2.11, 95% CI = 1.12–3.99) or high level of education (AOR = 1.97, 95% CI = 1.04–3.75) vs. a low level of education, ever tried to quit smoking vs. those who did not report a quit attempt (AOR = 8.81, 95% CI = 5.09–15.27), visiting a healthcare provider and receiving advice on how to quit vs. smokers who did not receive quitting advice from a healthcare provider (AOR = 3.78, 95% CI = 1.62–8.83), and being very worried (AOR = 7.35, 95% CI = 2.47–21.83) or a little worried/moderately worried (AOR = 3.11, 95% CI = 1.35–7.15) that smoking would damage their future health vs. those who were not worried at all. Gender, age, and nicotine dependence were not significantly associated with having quit intentions.

## 4. Discussion

Most Malaysian smokers (85.2%) reported having intentions to quit smoking, although less than half (43.6%) of them intended to quit within 6 months. This latter percentage (43.6%) is higher compared to the percentage in India (4%), Kenya (14%), Bangladesh (15%), Thailand (19%), China (28%), and Mauritius (28%) but lower than Brazil (48%) [40]. Five factors were significantly associated with quit intentions: ethnicity, level of education, prior quit attempt, receiving cessation advice from a healthcare provider, and worrying about the health effects of smoking. 

Malaysian smokers who reported visiting healthcare providers and receiving advice to quit smoking were more likely to have quit intentions (than smokers who did not receive quitting advice from a healthcare provider). This finding is consistent with other studies among smokers from Bangladesh, Taiwan, India, and the United Kingdom [27,41,42,43]. It is well-established in both high-income countries and LMICs that advice to quit from a physician or healthcare provider is a powerful motivator for quitting [27,41,42,43,44,45]. Thus, greater efforts need to be made to educate Malaysian doctors and healthcare providers regarding the harms of tobacco use and highlight the important role they play in providing quitting advice regularly to their patients who smoke. 

The study showed that respondents who reported worrying about smoking damaging their health in the future were more likely to have quit intentions, as is consistent with studies from other countries [10,21,26,27,28,31]. Hence, public education and cessation programs need to emphasize the harms of tobacco use to increase quit intentions and successful quitting. The government needs to fund and design strategic promotions using mass media channels to improve the public’s awareness of mQuit services and to encourage more smokers to use their smoking cessation services. Tailored mQuit programs strategically located in rural communities and in institutions, such as universities, schools, and private companies, should be developed. Malaysian government recently increased accessibility to mQuit services by creating a comprehensive smoking cessation system. The newly implemented system is incorporated in the “MySejahtera” application, which includes information dissemination, client registration, referral to private smoking cessation service providers, and follow-up monitoring by mQuit program partners. This allows smokers to contact a provider from the mQuit program located near them to receive cessation help [9]. The smoking cessation training programs certified by the mQuit program, such as the Smoking Cessation Organizing, Planning and Execution (SCOPE), and Certified Smoking Cessation Service Provider (CSCSP), should be continuously updated and disseminated to Malaysian healthcare providers to keep them current with the latest cessation information. This knowledge will enable healthcare professionals to provide effective smoking prevention and cessation intervention programs. 

It is also important for the “Speak out” program to emphasize the harms of smoking. Other cost-effective strategies used in other countries to increase awareness of the harms of tobacco use and increase quit intentions include increasing the size of the pictorial health warnings (PHWs) (i.e., more than 50% at the front) and regularly rotating them [46], and implementing plain packaging of cigarette packages [47].

Consistent with other studies [23,27,28,29,30,31], Malaysian smokers who reported past quit attempts were more likely to intend to quit (vs. those who did not report a past quit attempt). In a study of quitting among smokers in Canada, Chaiton et al. [48] found that those smokers who had made several quit attempts, including using quit aids, had a higher chance of successful quitting. This finding highlights the importance of strengthening cessation programs in Malaysia to encourage smokers to make multiple quit attempts until they successfully quit smoking. Additionally, guidelines of Article 14 of the WHO Framework Convention on Tobacco Control [49] recommend a broad range of cessation interventions, including population-based approaches that have wide reach (e.g., mass communication, brief advice, and quit-lines). These guidelines also recommend more intensive individual approaches, such as specialized treatment services, e.g., behavioral support and medications [49]. Malaysian smokers of Malay ethnicity and smokers with moderate or high levels of education were more likely to have a quit intention. Hence, targeted cessation programs (including mass media campaigns) for these smoking groups would be important.

The strengths of this study include the use of standardized measures associated with quit intentions, which have been used in 30 other ITC countries, and being one of the few nationally representative studies in Malaysia to examine the predictors of quit intentions. The limitations of this study include the use of self-reporting data, which may result in recall bias and social desirability bias. Additionally, the cross-sectional study design limits the ability to determine causality. There are many factors that are associated with quit intentions that we did not measure in our study, for example, depression and self-efficacy. Thus, there is need for further research to explore other factors that affect smokers’ quit intentions and successful quitting in different cultural contexts. Longitudinal studies are needed to determine causality.

## 5. Conclusions

The factors associated with quit intentions among Malaysian smokers are consistent with those identified in other countries. Understanding the predictors of quit intentions can strengthen existing cessation programs and guide the development of more effective cessation programs in Malaysia. For example, cessation awareness campaigns need to highlight the harms of tobacco use and consist of specific cessation messages for different groups of smokers, such as those having low levels of education or smokers from different ethnic groups. Additionally, doctors and other healthcare providers need to be consistently reminded to provide quitting advice to their patients who smoke and encourage them to make multiple quit attempts. These interventions could improve quit intentions and successful quitting among Malaysian smokers. These findings could also provide evidence for health policymakers from other countries in Southeast Asia.

## Figures and Tables

**Table 1 ijerph-19-03035-t001:** Characteristics of the sample of Malaysian smokers (unweighted).

Variable	Response	Frequency (N = 1047)	Percentage (%)
Gender	Male	944	90.2
	Female	103	9.8
Ethnicity	Malay	547	52.8
	Non-Malay	489	47.2
Age at recruitment	18–24 years old	144	13.8
	25–39 years old	635	60.6
	40–54 years old	223	21.3
	55 years and up	45	4.3
Education	Low (upper secondary and below)	352	33.8
	Moderate (Diploma Certificate)	300	28.8
	High (University and higher)	390	37.4

**Table 2 ijerph-19-03035-t002:** Smoking-related measures (weighted).

Variable	Response	Percent (%)	95% Confidence Intervals
Intentions to quit	No	14.8	12.1–17.9
	Yes	85.2	82.1–87.9
Time to quit intention	Within the next month	13.6	11.2–16.6
	Within the six months	28.4	25.0–32.1
	Beyond six months	43.2	39.2–47.2
	Not planning to quit	14.8	12.1–17.9
Ever tried to quit in past 12 months	No	14.1	11.8–16.8
	Yes	85.9	83.2–88.2
Healthcare professional visit and quit advice	Visited healthcare professional and received advice to quit	24.9	21.8–28.4
	Visited healthcare professional but did not receive advice to quit	11.5	9.1–14.4
	Did not visit a healthcare professional	63.6	59.8–67.3
Worried smoking damage health	Not worried at all	8.5	6.5–10.9
	A little worried/moderately worried	65.0	61.1–68.8
	Very worried	26.5	23.0–30.3
Spend too much on cigarettes	Disagree/strongly disagree/neutral	35.2	31.5–39.2
	Strongly agree/agree	64.8	60.8–68.5

**Table 3 ijerph-19-03035-t003:** Factors associated with intentions to quit smoking among Malaysian smokers.

Predictor	Category	Multivariable Model AOR (95% CI)
Gender	Male	Ref
	Female	1.03 (0.25–4.36)
Age at recruitment	18–24 years old	Ref
	25–39 years old	1.34 (0.57–3.18)
	40–54 years old	1.56 (0.58–4.16)
	55 years and up	0.40 (0.12–1.34)
Ethnicity	Non-Malay	Ref
	Malay	1.82 (1.03–3.20) *
Education	Low (upper secondary and below)	Ref
	Moderate (Diploma Certificate)	2.11 (1.12–3.99) *
	High (University and higher)	1.97 (1.04–3.75) *
Ever tried to quit in the past 12 months	No	Ref
	Yes	8.81 (5.09–15.27) *
Healthcare professional visit and quit advice	Did not visit a healthcare professional	Ref
	Visited healthcare professional and received advice to quit	3.78 (1.62–8.83) *
	Visited healthcare professional but did not receive advice to quit	1.26 (0.66–2.40)
Worried smoking damage health	Not worried at all	Ref
	A little worried/moderately worried	3.11 (1.35–7.15) *
	Very worried	7.35 (2.47–21.83) *
Spend too much on cigarettes	Disagree/strongly disagree/neutral	Ref
	Strongly agree/agree	1.47 (0.79–2.76)
Heaviness of Smoking Index (HSI) ^#^		0.86 (0.71–1.05)

Ref: reference category; *****
*p* < 0.05; ^#^ Heaviness of Smoking Index (HSI) was treated as a continuous variable. For smokers who had intentions to quit, the mean HSI was 2.14.

## Data Availability

In each country participating in the International Tobacco Control Policy Evaluation (ITC) Project, the data are jointly owned by the lead researcher(s) in that country and the ITC Project at the University of Waterloo. Data from the ITC Project are available to approved researchers 2 years after the date of issuance of cleaned data sets by the ITC Data Management Centre. Researchers interested in using ITC data are required to apply for approval by submitting an International Tobacco Control Data Repository (ITCDR) request application and subsequently to sign an ITCDR Data Usage Agreement. The criteria for data usage approval and the contents of the Data Usage Agreement are described online (http://www.itcproject.org) (accessed 25 February 2022).

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
