# Peer review of "Who Are More Likely to Have Quit Intentions among Malaysian Adult Smokers? Findings from the 2020 ITC Malaysia Survey"

_ijerph, 2022, doi:10.3390/ijerph19053035_

Round 1

Reviewer 1 Report

Congratulations for this excellent article. 

Please increase in the discussion section the limitations of the study.

Author Response

We thank the reviewer for their kind comment and suggestion.

We have added 2 sentences to expand the limitations section. The change is found in the Discussion section, sixth paragraph. It reads as follows (please see the text in next column in yellow highlights).

The limitations of this study include the use of self-reporting data which may result in recall bias and social desirability bias. Additionally, the cross-sectional study design limits the ability to determine causality. There are many factors that are associated with quit intentions that we did not measure in our study, for example, depression and self-efficacy. Thus, there is need for further research to explore other factors that affect smokers’ quit intentions and successful quitting in different cultural contexts. Longitudinal studies are also needed to determine causality.

Reviewer 2 Report

The authors as the purpose of this study indicated to analyze the factors associated with quit intentions among current cigarette smokers from Malaysia. The study was planned and carried out correctly. I have two comments to extend the analysis and discussion.

The statement that people who have reported attempts to quit smoking in the past still intend to quit may on the one hand indicate that the desire to quit smoking is a permanent feature, but on the other hand that their methods have been unsuccessful. With such a broad range of material available, the effectiveness of the various methods used to quit smoking could be analyzed and smokers could be offered the most effective methods of quitting smoking.  

The authors rightly point out the need the importance of tailoring cessation programs (including mass media education campaigns) to Malaysian smokers from other ethnicities and smokers with low levels of education. With such extensive material, it is possible to analyze the differences in the motivation to quit smoking between smokers from other ethnic groups and Malaysian smokers and smokers with low level of education vs smokers with higher level of education. Such information would make the postulate of tailoring the programs more realistic.

Author Response

Comment 1: Thank you for your kind comment and suggestion.

Our study showed that smokers who reported past quit attempts were more likely to have quit intentions than smokers who did not report quit intentions. Intentions to quit smoking is one of the strongest predictors of making future attempts to quit and of successful quitting. In order to do the analyses that you requested, i.e., to assess the most effective cessation methods that were used by smokers to successfully quit--- we would need to use longitudinal data that includes cohort respondents who will have reported that they had successfully quit (i.e., quitters). There is only one wave of data from the new cohort of the ITC MYS1 Survey which is cross-sectional, and it is not possible to address your comment. We hope to assess the cessation methods used by successful quitters once we have data from a follow up survey.

That said, if you are interested in cessation methods that have been used for past quit attempts by current smokers (not quitters), it is possible to use this cross-sectional data but one of our colleagues is currently writing a paper about this and we are happy to share it once it is published.

Comment 2: Thank you for your kind comment and suggestion.

It is possible to analyze the differences in reasons/motivation to quit smoking between smokers from other ethnic groups vs. Malay smokers and between smokers with low level of education vs smokers with high levels of education. However, this is beyond the scope of our paper which was mainly focusing on assessing the predictors of quit intentions. In order to address your comment, we would need to do a different analysis, i.e., make “motivation to quit” as the dependent variable instead of “having a quit intention” as the dependent variable as we have done in our paper. Thank you for the interesting idea. We will look into it in our next paper.

Reviewer 3 Report

The aim of the article “Who Are More Likely To Have Quit Intentions Among Malaysian Adult Smokers? Findings From the 2020 ITC Malaysia Survey” was, “to examine the factors associated with quit intentions among current cigarette smokers from Malaysia using data from the 2020 ITC Malaysia Wave 1 (MYS1) Survey”; this is an important paper; however, required some corrections.

Lines 121-123. Nicotine dependence was measured using the Heaviness of Smoking Index (HSI). Was this test validated in the Malaysian population? (PMID: 31519135).

Discussion section

Do these findings only apply to the Malaysian population? please, indicate the contribution of your study to other populations.

Author Response

We thank the reviewer for their kind comment.

Reviewer 4 Report

Thanks for the opportunity to review the manuscript. The current studied the factors associated with quitting smoking in Malaysian adult smokers using the latest national representative survey. A few modifications could help increase the soundness of the current study.  

Overall, a throughout grammar check is needed. There are some sentence fragments and typos.

Introduction:

Line 58-72: authors might consider introducing the trend of smoking prevalence during the same period to support the effectiveness of these programs.

Line 76-79 no details were needed about these listed theories and links needed for why these are important for conducting the current study.

Method:

The definition of a current smoker seems to miss the “current” part. The 100 cigarettes in a lifetime only measure whether they are established smokers.

The major issue is such a low response rate and a short period of data collection.

Authors might consider providing a detailed rationale for how these variables were selected and consideration of collinearity of these variables. Any sensitivity analyses (although results were not likely to change) were conducted to change the definition of the outcome variable (e.g., switching people intending to quit more than 6 months to no quit intention group).

Generally, bivariate analyses were not that interested if the multivariable regression results were included. Authors might consider deleting this part.  

Discussion:

Authors might consider adding more implications on how the government and other organizations could help not only the healthcare providers.

Author Response

Dear Reviewer,

Thank you for your kind suggestion.
